# VP-LLM: Text-Driven 3D Volume Completion with Large Language Models through Patchification

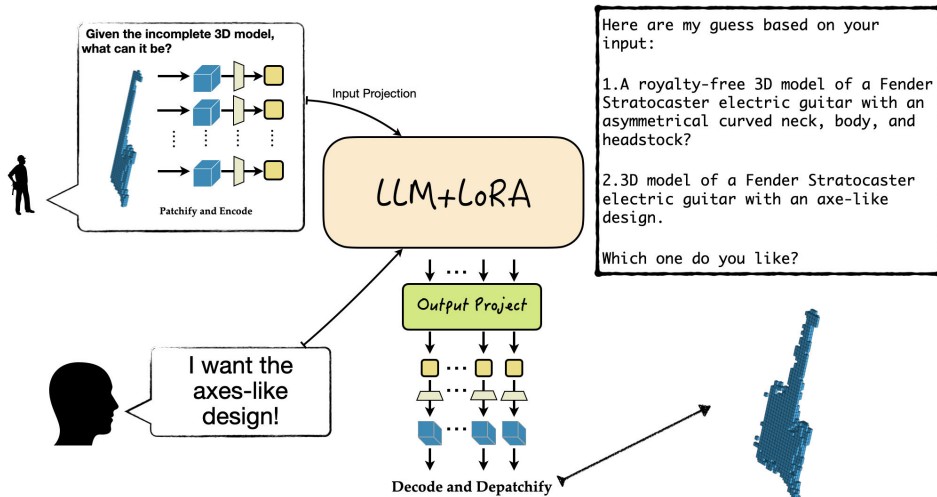

Figure 1: **Overview**. VP-LLM leverages the long-context comprehension capability of Large Language Models (LLMs) to process 3D models. It takes either incomplete or noisy 3D models along with textual instructions as input, and generate a complete model in an interactive way. This is achieved by segmenting the 3D object into patches and processing each independently.

## Abstract

3D completion represents a critical task within the vision industries. Traditional diffusion-based methodologies have achieved commendable performance; however, they are hindered by several issues. Firstly, these methods primarily depend on models such as CLIP or BERT to encode textual information, thereby making them incapable of supporting detailed and complex instructions. Moreover, their model sizes usually increase rapidly when the scene is larger or the voxel resolution is higher, making it impossible to scale up. Witnessing the significant advancements in multi-modal understanding capabilities facilitated by recent developments in large language models (LLMs), we introduce Volume Patch LLM (VP-LLM), designed to execute *user-friendly* conditional 3D completion and denoising using a token-based single-forward pass approach. To integrate a 3D model into the textual domain of the LLM, the incomplete 3D model is initially divided into smaller patches—a process we refer to as "patchification"—in a way that each patch can be independently encoded, analogous to the tokenization configuration utilized by LLMs. These encoded patches are subsequently concatenated with the encoded text prompt sequence and inputted into an LLM, which is fine-tuned to capture the relationships between these patch tokens while embedding semantic meanings into the 3D object. Our findings indicate a robust ability of LLMs to interpret complex text instructions and comprehend 3D objects, surpassing the quality of results produced by state-of-the-art diffusion-based 3D completion models, especially when complex text prompts are given.

# 1 INTRODUCTION

3D modeling serves as a pivotal component in a multitude of 3D vision applications including robotics and virtual reality, where the quality of 3D data critically influences model performance. Despite the advancement in 3D scanning technology, the raw data acquired are often noisy, clustered, and may contain large portions of missing data due to occlusion, complex real-world scenes and restricted camera angles, resulting in incomplete 3D acquisition. This necessitates robust pre-processing to recover or complete the 3D objects, which can enhance the efficiency of subsequent 3D vision tasks.

Current approaches for 3D shape completion typically operate on a depth map or partial point cloud, converting it into a voxel representation or sampling points to restore the original 3D objects. While Wu et al. (2020); Zhang et al. (2021) showcase advancements, they are confined to specific categories and lack the ability of cross-object generation. Although efforts have been made Yan et al. (2022); Yu et al. (2021); Wu et al. (2018); Wen et al. (2021) to create a unified model that handles multi-category 3D completion, these models often overlook the inclusion of textual input in guiding the completion process, leading to uncertainty when the input is ambiguous, as well as degradation of feasibility when given captions deviate from the training set. Consequently, methods are needed to generate a completed shape aligning precisely with the provided text description. Some attempts like Cheng et al. (2023); Kasten et al. (2023) mimic the 2D diffusion method or score distillation sampling (SDS) to incorporate text guidance in the 3D completion tasks, but they cannot be precisely controlled when the description is complicated, and are very time-consuming to generate the results.

To this end, we propose Volume Patch Large Language Model (VP-LLM), which achieves 3D completion with precise textual control. Inspired by the recent progress in 3D multi-modality models (Yin et al., 2023; Chen et al., 2023b; Wang et al., 2023c), we believe that Large Language Models (LLMs) can underpin our approach by decoding the complex associations between 3D structures and textual descriptions. LLMs, pretrained on large-scale text datasets, have the capability to process long sequences and comprehend complex human languages, while 3D models represented by voxel grids can be straightforwardly converted into a one-dimensional format through flattening. Therefore, we investigate how to enable LLMs to understand a 3D model by decoding complex correlations between 3D structures and textual descriptions, or "translating" it into a "sentence".

For seamless incorporation of 3D data into the LLM tokenization framework, 3D models are initially segmented into smaller patches, facilitating independent encoding and decoding. Different from most previous methods that manage the 3D object as a unified, this idea of patchification is more scalable and extendable. The patchified 3D voxel volume can be processed as a sequence and fed into the LLM with the textual description after alignment. The LLM can fuse the 3D and textural features into its hidden latents, which are finally decoded into complete 3D models.

Our whole pipeline is presented in Fig. 1. The 3D volumes are first patchified into individual patches and processed by a patch-wise Variational Autoencoder (VAE) to encode individually. The encoded patches are then projected and concatenated with user-specified text conditions to the LLM. Finally, the output projection layer extracts the features generated by the LLM and lets the VAE decode back each patch individually.

In summary, the major contributions of our papers are:

1. We proposed a *patchification* method, which enables a scalable integration of 3D volumes into the LLM, which is akin to LLM's tokens, solving the difficulty in handling high-resolution voxel grids faced by existing works.

2. VP-LLM is the first work leveraging *LLM* to achieve 3D completion with *precise* text-control, which outperforms existing state-of-the-art text-conditioned 3D completion works.

3. Our work serves as an interactive *unified* agent that performs 3D understanding, completion and denoising for multiple categories with *detailed* text control.

Thus, this experimental paper offers insights and lessons learned, providing the first LLM solution to text-guided 3D object completion. Codes and data will be made public upon the paper's acceptance.

## 2 RELATED WORKS

### 2.1 MULTIMODALITY LARGE LANGUAGE MODELS

The advent of Large Language Models (LLMs) has significantly accelerated advancements in natural language processing. Several studies like Jiang et al. (2023b); Touvron et al. (2023); Team et al. (2024) have demonstrated the capabilities of LLM in comprehending long contexts, ensuring scalability and adaptability, and facilitating the understanding and generation of natural language. Benefiting from these advantages of LLM, many works have already employed LLM across different modalities, including images (Wang et al., 2023b; OpenAI et al., 2024; Alayrac et al., 2022), motion (Jiang et al., 2023c), and video (Zhang et al., 2023; Li et al., 2024). Recently, several works combined LLM with 3D data. For example, Yin et al. (2023) leverages a 3D-aware VQ-VAE (van den Oord et al., 2018) and integrates its codebook into the LLM's vocabulary, enabling the LLM to generate and understand 3D objects. But the codebook size may bottleneck the capability of LLM to tackle 3D objects with more complex and various structures. LLM-Grounder (Yang et al., 2023) carefully designs LLM prompt to translate the instructions into regular sub-tasks and instructs some pre-trained 3D grounders Kerr et al. (2023); Peng et al. (2023); Qi et al. (2024); Guo et al. (2023); Hong et al. (2023) for 3D reasoning, where 3D models are not integrated into the LLM. Octavius (Chen et al., 2023b) adopts the object detector to first discover candidate regions, followed by the application of pre-trained point cloud encoders for extracting features at the instance level. These features are then aggregated and mapped into an LLM for diverse 3D understanding tasks. However, this process reduces the entire 3D model to a single feature, thereby omitting crucial detailed information. In contrast, VP-LLMemploys a VAE (Kingma & Welling, 2013) combined with projection layers, capable of effectively aligning the 3D latent space with the LLM's text space, thus enhancing the model's generalizability, especially for out-of-distribution data.

### 2.2 TEXT-TO-3D GENERATION

Prior to the era of machine learning, primitive works attempted to retrieve 3D assets from large databases, such as Chang et al. (2014; 2015a). With the rise of GANs (Goodfellow et al., 2014), attempts such as Text2Shape (Chen et al., 2019) started to dominate the 3D generation field. Recently, due to promising advancements in text-to-image generation, research focus on text-control 3D generation has shifted to diffusion model (Ho et al., 2020). Some works Liu et al. (2023a); Sanghi et al. (2022); Jain et al. (2022) adopt CLIP (Radford et al., 2021) to align the rendered images with the input text, thus ensuring the semantic meaning of the 3D model, while others like Poole et al. (2022); Wang et al. (2024; 2023a); Chen et al. (2023a); Lorraine et al. (2023); Babu et al. (2023) leverage pre-trained 2D diffusion models to provide text control and score distillation sampling to improve the 3D consistency. Although text-to-3D generation serves as an inspiration for text-guided 3D completion, none of the existing methods employ large language models for 3D-text interaction to guide the completion results.

### 2.3 3D COMPLETION

3D completion is a crucial process in various industries, enabling accurate and efficient design and production, and enhancing the overall quality of products and projects. Early works such as Choy et al. (2016); Dai et al. (2017); Girdhar et al. (2016); Han et al. (2017); Stutz & Geiger (2018; 2020); Wu et al. (2015) that use 3D convolutions with structured representation require high memory usage and compute. Followed by Yuan et al. (2018), many works An et al. (2024); Tchapmi et al. (2019); Yu et al. (2022); Wu et al. (2020) that adopt point clouds as 3D representation for shape completion were proposed. For example, An et al. (2024); Tchapmi et al. (2019); Yu et al. (2022) generate the final shape in an auto-regressive manner, while Wu et al. (2020) uses GANs (Goodfellow et al., 2014) to complete the model. But none of these offer satisfactory user control. With the introduction of DDPM (Ho et al., 2020), works such as Li et al. (2023b); Liu et al. (2023b); Luo & Hu (2021); Vahdat et al. (2022); Wu et al. (2024); Rao et al. (2022); Chu et al. (2024); Zhou et al. (2021); Li et al. (2023a) have advanced the 3D shape completion pipelines conditioned on labels. Notably, Cheng et al. (2023) adopts a 3D diffusion model with VAE to achieve 3D completion with multi-type control, and Kasten et al. (2024) uses score distillation to perform test-time optimization. However, neither of them can perform completion tasks at a larger scale where model details are required, nor even

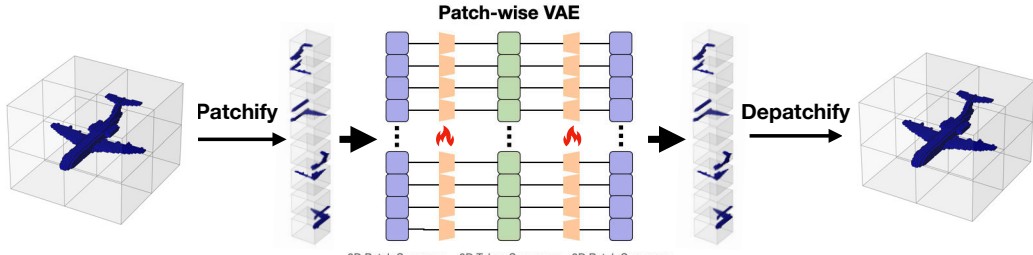

Figure 2: **Patchification**: given a 3D object, we first fit it into a voxel grid and then divide it into a sequence of small patches. Next, we utilize a patch-wise Variational Autoencoder (VAE) to extract the features of each patch individually and then reconstruct it back. It is important to note that only one VAE is trained for all the patches throughout the entire dataset, making our method a scalable approach.

generate satisfactory results without tedious denoising steps. Thus, scalability, speed and level of control are still issues to be addressed, providing strong motivation for our work.

## 3 METHODOLOGY

Given an incomplete 3D model and user-supplied textual description of the target 3D model, our model aims to recover the underlying 3D model aligned with the input text. First, the incomplete 3D model undergoes *patchification*, where it is split into small patches, and each patch is independently encoded by our Variational Autoencoder (VAE). Next, a shared-weight linear layer maps the patch features to the embedding space of the LLM, which are then combined with the textual description and input into the LLM. The LLM, concatenated with our specially-designed output projection layer, generates the features of patches at all positions, allowing for the separate decoding and subsequent assembly, or de-patchification, to the underlying complete 3D model.

### 3.1 PATCHIFICATION

The first step of our method is to divide the 3D models into small patches, dubbed patchification. Figure 2 demonstrates the process of patchification. For a 3D object represented in voxel $V \in \{0,1\}^{H \times W \times D}$, $V(x,y,z) = 1$ if the position $x, y, z$ is occupied and 0 otherwise. Patchification uniformly partitions the 3D voxel volume into $p$ small patches of the same size, each containing a local region of the entire object. For each patch $P_{i,j,k} \in \{0,1\}^{h \times w \times d}$, the coordinate for position $(x,y,z), 0 \le x \le h, 0 \le y \le w, 0 \le z \le d$ is:

$$P_{i,j,k}(x,y,z) = V(i \cdot h + x, \ j \cdot w + y, \ k \cdot d + z). \tag{1}$$

Thus, $p = \lfloor H/h \rfloor \cdot \lfloor W/w \rfloor \cdot \lfloor D/d \rfloor$. In our experiments, we set $H = W = D = 64$ and $h = w = d = 8$.

After patchification, a patch Variational Autoencoder (VAE) is adopted to extract the feature for each patch independently. Our patch VAE consists of an encoder $\mathbf{E}$ and a decoder $\mathbf{D}$, where $\mathbf{E}$ encodes a patch into a Gaussian distribution $\mathcal{N}(\mu, \sigma)$ where $\mu$ and $\sigma$ are mean and variance, respectively, and $\mathbf{D}$ recovers the original patch from this distribution. The VAE training loss for a single patch $P$ is defined as

$$\mathcal{L}_{VAE}(P) = \mathcal{L}_{BCE}\left(P, \ \mathbf{D}(\mathbf{E}(P))\right) + \beta \mathcal{L}_{kld}(\mathbf{E}(P)), \tag{2}$$

where $\mathcal{L}_{BCE}$ is the binary cross entropy loss, $\mathcal{L}_{kld}$ is the KL-divergence and $\beta$ is a hyperparameter.

Benefiting from this patchification structure that allows for independent encoding and decoding of each patch, VP-LLM ensures that when the completions of certain patches are undesired, they do not affect the performance of other well-performed patches, solving the problems faced by previous works that encode or decode the entire scene collectively.

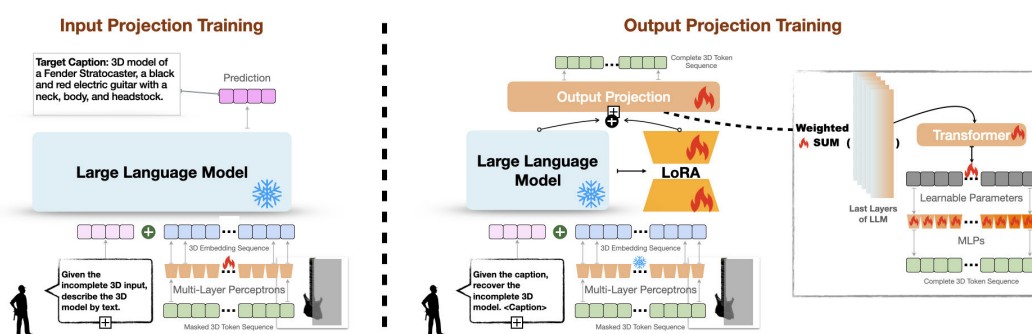

Figure 3: The training process of the **input projection** (left) and **output projection** (right). During the input projection training, a single share-weighted MLP maps the masked or noisy 3D tokens encoded by our patch-wise VAE to the embedding space of the LLM. After wrapping the prompt with the 3D tokens as input and feeding them to the LLM, we back-propagate the loss calculated between the ground-truth caption and the LLM's prediction, enabling the LLM to learn to generate captions that accurately describe the 3D object from the input patches. For the output projection, we freeze the input projection layer and train the output projection layer, while also fine-tuning the LLM with LoRA. The output projection layer comprises a Transformer and a cluster of MLPs, such that after passing the Transformer, every 3D token is processed independently with an MLP.

## 3.2 MASK STRATEGY

To enhance the understanding of incomplete 3D models, we designed three different strategies to mask out different parts of the original 3D input, aiming to mimic the possible user inputs during the inference stage. Specifically, the following three strategies will be applied randomly with the same possibility:

1. *Random Mask*: Given the input 3D model in $p$ patches, we randomly set $m_r \cdot p$ patches to $0$ (unoccupied), where $m_r$ is a mask ratio sampled within a pre-defined range;

2. *Plane Mask*: Given the input 3D model represented in voxel $V(x, y, z)$, we first project the model onto $x$-axis and find the first and last occupied voxels, denoted as $x_1$ and $x_2$, along $x$-axis. Next, a plane parallel to $yOz$ is sampled with $x$-coordinate between $[x_1, x_2]$. Intuitively, such a plane cuts the 3D model into two parts, and we then discard one of them by setting all voxels to be $0$ (unoccupied) to simulate large portions of missing data in real capture;

3. *Random Noise*: Since real-world 3D models usually contain noises and artifacts, we mimic this situation by randomly inverting voxel occupancy (setting occupied voxels to be unoccupied, and vice versa). The noise level is also sampled in a pre-defined range indicating how many voxels to invert.

## 3.3 INPUT PROJECTION LAYER TRAINING

The input project layer, which can map the VAE latent space into the LLM input embedding space, is a single linear model operated on each VAE latent patch. After patchifying and encoding the incomplete 3D model, each patch, represented by its respective $\mu$ and $\sigma$, is first reparameterized into a single feature $f$ by sampling from the Gaussian distribution $\mathcal{N}(\mu, \sigma)$. The feature (or each encoded patch) going through the input projection layer becomes a 3D token, which then can be understood by the LLM. To train the input projection layer, the LLM is instructed to predict the caption of the incomplete 3D model, given the prompt "Given the incomplete 3D input, describe the 3D model by text." We use the training loss in Radford et al. (2019) which is the negative log-likelihood on the caption. We use the training loss from the LLM to supervise this stage of training.

### 3.4 OUTPUT PROJECTION LAYER TRAINING

The LLM, with the output projection layer appended, takes as input the tokenized sequence of the underlying incomplete 3D model, the user-supplied caption of the complete model as well as the instructions for completion, and outputs the complete 3D model aligned with the textual description. To be specific, we formulate the prompt for LLM as "`Given the caption, recover the incomplete 3D model, <tokenized incomplete 3D sequence>, <Caption>`". The output projection layer architecture employs a transformer consisting of a 2-layer encoder and a 2-layer decoder, translating the LLM hidden states into a sequence of separable latent codes. We observe that to sufficiently explore the highly fused information in the LLM, more layers of hidden states are necessary. Thus, we select 5 layers in our experiments, which balances the amount of information with computational complexity. To ensure the length of the generated sequence, we set an additional learnable token sequence as the target. We then utilize Multi-layer Perceptrons (MLPs) to individually map each token to the desired 3D token. The final result is obtained through the concatenation of the mapped tokens.

During training, the input projection layer is frozen, while the LLM is finetuned with LoRA Hu et al. (2021), while the output projection layer is trained from scratch. We use mean-squared error (MSE) loss between the output projection layer output and the VAE latent of ground-truth 3D model patches to update our model.

### 3.5 INTERACTIVE WORKING FLOW WITH DETAILED INSTRUCTION

In order to demonstrate the 3D understanding ability of our LLM, we also provide an interactive completion and denoising interface, where the user initially inputs the incomplete model to VP-LLM, and our LLM, combined with our trained input projection layer, can provide potential completion options. The user is afforded the flexibility to either select from these options or input their own control instructions. Ultimately, they obtain the desired completed results, which is the output of our entire pipeline.

As illustrated in Fig. 1, after the user inputs half of a guitar, our LLM responds with two options to either complete as a guitar with *asymmetrical* body or *axe-like* body. With the user choosing the second one, the model outputs the corresponding results. We will provide more examples demonstrating our model's detailed controllability in the experiment section, where our model can distinguish between subtle differences in text instructions and understand them precisely.

## 4 EXPERIMENTS

### 4.1 DATASET

We train our model on a subset of ShapeNet (Chang et al., 2015b) dataset, comprising over 3000 objects. To obtain the detailed textual description of 3D models in human languages, we adopt Cap3D (Luo et al., 2024), which leverages BLIP to predict and GPT-4 to refine captions for 3D models.In our experiments, the resolutions of the 3D voxels and patches are respectively set as $64 \times 64 \times 64$ and $8 \times 8 \times 8$, while we explore the capability of the model to handle higher resolution formats in Sec. 5.2.

To improve the robustness of our model, we apply data augmentation during training. For the 3D models, we rotate them along one random axis with an arbitrary angle making the order of sequence different, while for the captions of the 3D model, we adjust the GPT configurations in Cap3D. Only 3D data augmentation is used in input projection training, while both 3D data and caption augmentation are used in output projection training. More details of the data augmentation can be found in Appendix C.

### 4.2 COMPARISON ON COMPLETION AND DENOISING TASKS

We compare our model with SDFusion (Cheng et al., 2023) and 3DQD (Li et al., 2023a), which are two state-of-the-art diffusion-based methods on conditional 3D completion tasks. SDFusion

Table 1: **Quantitative results compared with SDFusion and 3DQD**. We can observe that our method consistently hits the lowest (best) Chamfer Distance (CD) and the highest (best) CLIP-s score compared with SDFusion (text-conditioned completion) and 3DQD (label-conditioned completion). Moreover, our method is capable of denoising extremely noisy 3D inputs, while the baselines cannot accomplish the task.

| Methods | Seg 20% | | Seg 50% | | Seg 80% | | Noise 1% | | Noise 2% | |
|---------|---------|---------|---------|---------|---------|---------|---------|---------|---------|---------|
| | CD.↓ | CLIP-s.↑ | CD.↓ | CLIP-s.↑ | CD.↓ | CLIP-s.↑ | CD.↓ | CLIP-s.↑ | CD.↓ | CLIP-s.↑ |
| Ours | **10.96** | **27.80%** | **11.37** | **27.71**% | **17.42** | **25.17%** | 16.03 | 23.92% | 34.44 | 23.07% |
| SDFusion | 95.44 | 26.66% | 137.31 | 26.20% | 235.98 | 22.22% | – | | – | |
| 3DQD | 172.89 | 22.63% | 170.20 | 22.62% | 196.12 | 22.62% | – | | – | |

Table 2: **Comparison of our method with SDFusion and 3DQD, on Airplane dataset.** We can clearly see our method outperforms their methods when the input is segmented by a plane and performs reasonably well when the input is added with noises. Since the two baselines cannot work on noisy inputs, "N/A" is placed instead.

| Ground Truth | Masked/Noisy | Ours | SDFusion | 3DQD |

**Seg 20%**

*"3D model of a Boeing 747-400 featuring a spherical fuselage shell, truncated oblate wings, and made of aluminum and steel."*

**Seg 50%**

*"3D model of a toy airplane featuring a wing, fuselage, tail, rudder, and propeller, available in 3ds Max, OBJ, FBX, and C4D formats."*

**Seg 80%**

*"Royalty-free 3D model of a stealth fighter jet, featuring a delta wing with horizontal and vertical stabilizers"*

**Noise 1%**

*"3D model of a Saber fighter jet featuring detailed wings, fuselage, and multiple landing gears, compatible with 3ds Max, Maya, Blender, and other 3D software."*

**Noise 2%**

*"Royalty-free 3D model of a Boeing 747-400 jumbo jet."*

accepts multi-modality input for shape completion, so in our case, we only enable text-conditioned completion and enforce no image condition. The aforementioned baseline models are all trained on either a subset or the full ShapeNet dataset. This ensures the fairness of our comparison, as our dataset constitutes a subset of the training data used by all the baseline models.

The quantitative comparison results are shown in Tab. 1. Following Cui et al. (2024); Li et al. (2023a), we use Chamfer Distance (CD) and CLIP-s score for evaluation. For Chamfer Distance, we transform our volume representation into point clouds and use the coordinates in the voxel grids. For the CLIP-s score, we render 20 different-view 2D images around each volume and take the maximums among the CLIP feature scores between the images and the textual description. We test the performance of the models under various circumstances, by segmenting the different fractions of the object and adding random noise to the objects. We also provide visualization results in Tab. 2 and Tab. 3.

Table 3: **Comparison of our method with SDFusion and 3DQD, on Car dataset.** We can clearly see our method outperforms their methods when the input is segmented by a plane and performs reasonably well when the input is added with noises. Since the two baselines cannot work on noisy inputs, "N/A" is placed instead.

| | Ground Truth | Masked/Noisy | Ours | SDFusion | 3DQD |
|---|---|---|---|---|---|
| Seg 20% | | | | | |

*"3D model of a Chevrolet Tahoe pickup truck in black and red, available in multiple formats including OBJ and FBX."*

| Seg 50% | | | | | |

*"Royalty-free 3D model of a Mercedes-Benz SLK sports car"*

| Seg 80% | | | | | |

*"3D model of a muscle car with a hood, fenders, and a hood scoop."*

| Noise 1% | | | | N/A | N/A |

*"3D model of a yellow Dodge Viper SRT10 sports car."*

| Noise 2% | | | | N/A | N/A |

*"A 3D model of a police car, available royalty-free, with previews from different angles."*

It is worth noticing that 3DQD is a label-conditional completion models that do not include detailed text control during inference, leading to large variances in the prediction results. Though SDFusion contains text control in completion, it adopts BERT (Devlin et al., 2018) for text encoding, thereby suffering from complex text understanding.

More qualitative results are presented in Appendix F.

### 4.3 COMPARISON ON COMPLETION TASK WITH PRECISE TEXT CONTROL

In Fig. 4, we present VP-LLM's detailed text-control ability here. Specifically, when we alter the text prompt in a subtle manner, our model can capture the minor difference in the semantic meaning of prompts, thereby generating a different result. For example, when we instruct the model to generate an SUV with either a *roof-mounted gun* or a *solar panel on the roof*, VP-LLM can successfully

Table 4: **Comparison of our method with SDFusion when** *subtle differences* **in text instructions present.** We can clearly see our method is able to generate 3D objects that obey *detailed, precise* text prompts, while SDFusion fails to distinguish the subtle differences and instead, generates relatively general objects, even though the partial 3D input may contain some clues about the differences.

| Ground Truth | Masked (50%) | Ours | SDFusion |
| --- | --- | --- | --- |

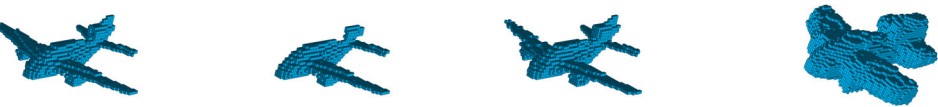

*"3D model of a Boeing 747-400 aircraft, showcasing detailed structure and geometry of the wing and fuselage."*

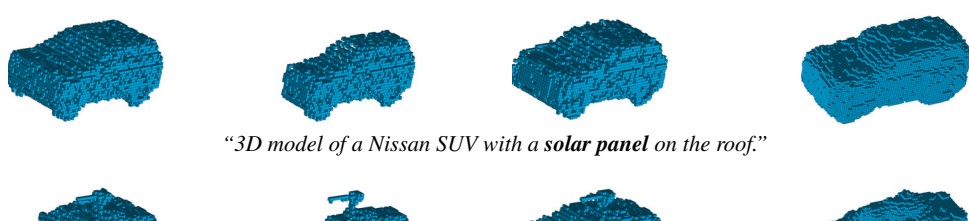

*"3D model of a Boeing 747-400 aircraft **with wings perpendicular to the aircraft body**, showcasing detailed structure and geometry of the wing and fuselage."*

*"3D model of a Nissan SUV with a **solar panel** on the roof."*

*"3D model of a Nissan SUV with a **roof-mounted gun**.*
*with horizontal and vertical stabilizers"*

Table 5: **Comparison using different LLM structures.** We utilize two LLMs, namely Mistral-7B and Gemma-2B, to train our whole pipeline with the same dataset as before. For comparison purposes, we demonstrate the Chamfer Distance (CD) and CLIP-s score on data containing 1% noise and 20% mask.

| Methods | Seg 20% | | Noise 1% | |
| --- | --- | --- | --- | --- |
| | CD.↓ | CLIP-s.↑ | CD.↓ | CLIP-s.↑ |
| Mistral-7B | **10.96** | 27.80% | 16.03 | 23.92% |
| Gemma-2B | 11.19 | **28.08%** | **10.64** | **26.34%** |

Table 6: **Comparison on different voxel volume resolutions.** We selected two different voxel volume resolutions, namely $H = W = D = 64$ and $H = W = D = 72$. We can see the two resolutions have comparable results. Thanks to our patchification method that enables each patch to be processed and generated independently, our method perfectly scales when the resolution is larger.

| Methods | Seg 20% | | Noise 1% | |
| --- | --- | --- | --- | --- |
| | CD.↓ | CLIP-s.↑ | CD.↓ | CLIP-s.↑ |
| Resolution $64^3$ | 10.96 | 27.80% | 16.03 | 23.92% |
| Resolution $72^3$ | 12.33 | 27.38% | 12.11 | 27.05% |

generate reasonable objects with correct semantic meaning, while SDFusion tends to ignore the difference in between.

## 5 ABLATION STUDY

### 5.1 COMPARISON OF DIFFERENT LLM ARCHITECTURES

LLM is one of the most important components in our model, whose ability to capture multi-modal semantic information and token relations may greatly affect the performance of the entire pipeline. To investigate the effect of different LLMs on the performance of our model, we re-trained our method on Mistral-7B (Jiang et al., 2023a) and Gemma-2B (Team et al., 2024). To ensure a fair comparison, we keep the LoRA ranks for the two models to be the same. From Tab. 5.1, we find that under this setting, the two LLMs demonstrate similar results on mask completion, while the performance of Gemma-2B is better than Mistral-7B on denoising tasks while slightly worse in completion from masked inputs. Higher performance can be expected if larger LLM models or higher LoRA ranks are deployed.

### 5.2 SCALABILITY ON HIGHER RESOLUTION VOXEL VOLUMES

Our VAE encodes and decodes each patch of the 3D model individually, thus enabling our model to scale to higher voxel resolutions. To demonstrate the scalability, we have expanded upon our previous experiments by setting $H = W = D = 64$, and further increasing the voxel resolution to $H = W = D = 72$, while maintaining the patch size at $8$. By fixing the patch size, we can ensure that the fine details of the data remain consistent as we increase the input scale. After this operation, the sequence length of each 3D object will increase from $512$ to $729$ (around $42\%$ increase). We can see from Tab. 6 that the two resolutions have comparable results, indicating our method can successfully generalize to higher voxel volume resolutions.

On the other hand, we would like to point out that the resolution $64^3$ is already the highest among voxel grid works. The most recent top conference works use much smaller resolutions, such as Rao et al. (2022) uses $32^3$, and Liu & Liu (2021); Tu et al. (2023) both use $40^3$. Moreover, considering most of the modern small LLM models (even 2B models like Gemma-2B) can handle at least 4K-8K context length, our method can perfectly handle resolutions to $128^3$ (which requires 4K context length). This is way more than common requirements and it does not make sense to demand model to handle even higher resolutions.

## 6 LIMITATION

Our method has been proven effective on small LLMs. However, due to the limitation of computational resources, LLMs with stronger capability to understand long sequences, such as 70B or larger scale models, are not employed in our model. Thus, huge potential of our method is still left to probe. Additionally, though we verify the effectiveness of our method by patchification on voxel volumes, which can be intuitively extended to point clouds and SDFs, it is still very hard to employ it on those nascent 3D representations like NeRF (Mildenhall et al., 2020) and 3D Gaussian Splatting (Kerbl et al., 2023) that encoded 3D in an implicit (MLP weights for NeRF and Gaussians for 3DGS). More investigations on how to patchify such representations are left in future works.

## 7 CONCLUSION

In this paper, we present VP-LLM, which combines text and 3D through LLMs to achieve text-guided 3D completion with detailed, precise semantics of texts captured. We introduce a novel approach called patchification to incorporate 3D models into LLMs, and adopt a two-stage training process that allows LLMs to understand input incomplete 3D models and generate entire 3D models. Such an approach also allows an independent encoding and decoding process, thereby ensuring the scalability of our method. Experiments on the ShapeNet dataset validate that our method surpasses the state-of-the-art methods in the 3D completion task. Moreover, our model can achieve satisfactory results from noisy 3D inputs with an interaction interface, a practical issue in real 3D data capture.

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

## A   IMPLEMENTATION DETAILS

**Dataset preparation**   We randomly selected 3130 data from ShapeNetCore Chang et al. (2015b), and randomly split data of each category into 90% training data and 10% testing data.

**Model structure**   We present the hyperparameters in Tab. 7. Those values can determine the detailed structures of each component in our pipeline.

| HYPERPARAMETER | VALUE |
| --- | --- |
| VAE Encoder Convolution Layer Num | 2 |
| VAE Decoder Convolution Layer Num | 2 |
| VAE Hidden Dimension | 64 |
| VAE Latent Size | 128 |
| LoRA Rank | 32 |
| LoRA Alpha | 32 |
| LoRA Dropout | 0.05 |
| LLM Type | Mistral-7B |
| Output Projection Transformer Encoder Layer Num | 2 |
| Output Projection Transformer Decoder Layer Num | 2 |
| Output Projection Transformer Feedforward Dimension | 2048 |
| Output Projection Transformer Num Heads | 4 |

Table 7: Hyperparameters used to configure model structure.

**Training configuration**   We train our patch VAE on 2 RTX 4090 cards for 100 epochs and batch size using ShapeNet voxels volumes with resolution $64^3$ and adopt the same VAE for voxels with different resolutions. This training takes approximately 2 hours. The model is trained with AdamW Loshchilov & Hutter (2017) optimizer with a learning rate $3e^{-4}$.

Our input projection and output projection models are trained on 8 RTX 6000 Ada GPU cards until converge, for around 100 and 500 epochs respectively. For Mistral-7B, these processes take around 1 hour and 18 hours, respectively, while for Gemma-2B, the numbers go down to 20 minutes and 8 hours. Both stages are trained with AdamW optimizer as well, with input projection training using a learning rate of $3e^{-4}$ and output projection training using $5e^{-4}$ or $5e^{-5}$, depending on the LLM size.

## B   EXAMPLES OF GROUND-TRUTH AND PREDICTED CAPTIONS

Here we present some examples of ground-truth captions and the captions predicted by our LLM model during input projection layer training. We notice that the captions are not perfect, while they provide adequate semantic meanings.

PREDICTED CAPTION 1: 3AD model of a ", featuring a exterior such as wings, fuselage, and, andinglets, and, and, andilerons, and flaps" with for 3 and7-400 and 747-800 variants.",

GROUND-TRUTH CAPTION 1: "3D model of Boeing aircraft, featuring detailed components such as wings, fuselage, tail, winglets, rudder, elevators, ailerons, and flaps, available in both 747-400 and 737-800 variants."

PREDICTED CAPTION 2: 3D model of a rectangular 737- featuring a fuselage body with a wings, and tail tail tail, and, and, andilerons, and a gear.,

GROUND-TRUTH CAPTION 2: 3D model of a Boeing 747, featuring a cylindrical fuselage, elliptical wings, a truncated cone tail, rudder, elevators, ailerons, and landing gear.

PREDICTED CAPTION 3: 3D model of of a guitars Ghostcar aircraft jets, including a-A-18E F-14., with for download3ds Max. OBJ.,

GROUND-TRUTH CAPTION 3: 3D model collection of various Phantom and Super Hornet fighter jets, including F/A-18 and F-16 variants, available for 3ds Max and Maya.

PREDICTED CAPTION 4: 3D model of a rectangular 737-800 aircraft a fuselage dome, with a conelate spher, and a- a. steel.,

GROUND-TRUTH CAPTION 4: 3D model of a Boeing 747-400 featuring a spherical fuselage shell, truncated oblate wings, and made of aluminum and steel.

PREDICTED CAPTION 5: 3 with a cylindrical, a, and, and, and,, and landing landing gear.,

GROUND-TRUTH CAPTION 5: A spaceship featuring a wing, fuselage, tail, propeller, rotor blade, and retractable landing gear.

PREDICTED CAPTION 6: 3D model of a electric with a, a, and, and, and, andilerons, and gear, and a.,

GROUND-TRUTH CAPTION 6: 3D model of an aircraft featuring wings, fuselage, tail, rudder, elevators, ailerons, landing gear, and propeller.

PREDICTED CAPTION 7: 3Aalty-free 3D model of a female 737-400 aircraft featuring a exterior such as a fuselage, fuselage, and tail.",

GROUND-TRUTH CAPTION 7: "Royalty-free 3D model of a Boeing 747-400, featuring detailed components such as a wing, fuselage, and tail."

PREDICTED CAPTION 8: 3D model of a rectangularliner with a fuselage wing, a fuselage, and, and, andders, and, and gear, and landing.,

GROUND-TRUTH CAPTION 8: 3D model of a jet plane featuring a delta wing, triangular fuselage, tail, fin, rudders, propeller, landing gear, and hull.

## C  ILLUSTRATIONS OF DATA AUGMENTATION

**3D Data Augmentation**  During training, we performed 3D augmentation by randomly rotating each 3D object with a different angle with respect to either $x$, $y$, or $z$ axis. Figure 4 visualizes this result.

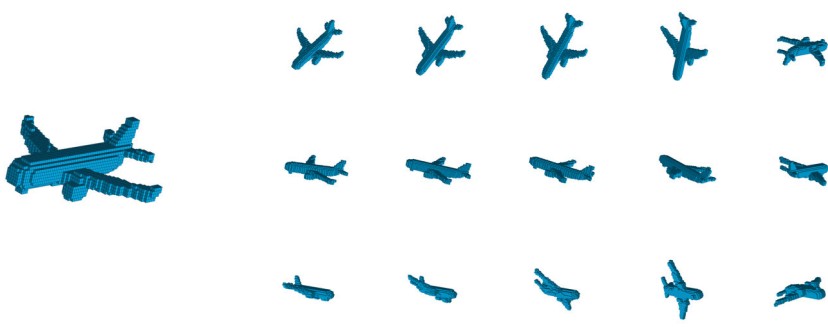

Figure 4: 3D data augmentation example result of an airplane.

**Caption Augmentation**  During training, we leveraged Cap3D Luo et al. (2024) to generate ground-truth captions for every 3D model. To perform caption augmentation, we run Cap3D for three times,

using GPT-4-Turbo, GPT-4-Turbo with another seed, and ChatGPT (GPT-3.5). Here we present the captions generated by them.

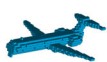 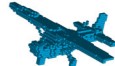 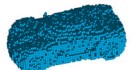 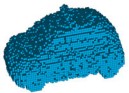

Figure 5: From left to right: Object 1, 2, 3, 4.

Captions for Object 1:

GPT-4 SEED 1: Royalty-free 3D model of a Boeing 747-400 featuring detailed components including a cylindrical fuselage, delta wings, tail, rudder, elevators, and ailerons.

GPT-4 SEED 2: Royalty-free 3D model of a Boeing 747-400 featuring a cylindrical fuselage, delta wings, a tail, rudder, elevators, and ailerons, representing a four-engine jet airliner.

CHATGPT: Boeing 747-400 3D model featuring a fuselage, wings, and tail, a jumbo jet with a delta wing design and four engines.

Captions for Object 2:

GPT-4 SEED 1: A 3D model of a two-seater, single-engine RC airplane featuring a four-bladed, fixed-pitch propeller, retractable tricycle landing gear, and control surfaces including ailerons, rudder, and elevator.

GPT-4 SEED 2: A 3D model of a small, two-seater, single-engine RC airplane featuring a retractable tricycle landing gear, a fixed-pitch, four-bladed propeller, and control surfaces including wings, fuselage, tail, rudder, elevator, and ailerons.

CHATGPT: 3D model of a two-seater single-engine airplane with retractable landing gear and a four-bladed propeller.

Captions for Object 3:

GPT-4 SEED 1: Royalty-free 3D model of a blue McLaren MP4-12C sports car with polygonal geometry.

GPT-4 SEED 2: Royalty-free 3D model of a McLaren MP4-12C sports car with polygonal geometry.

CHATGPT: 3D model of a McLaren MP4-12C sports car.

Captions for Object 4:

GPT-4 SEED 1: 3D model of a police car, available royalty-free, featuring detailed polygonal geometry.

GPT-4 SEED 2: 3D model of a police car, featuring detailed polygonal vertices and edges, available royalty-free.

CHATGPT: A detailed 3D model of a police car.

# D   UNLOCK BETTER QUALITY VIA ITERATIVE COMPLETION

We also found that our model is able to refine the results without further training, by directly passing the output from the last step and the caption into our model. We show this using the denoising

example as the changes are more obvious. After applying the denoising twice, we observe that the quality in Fig. 6 has significantly increased.

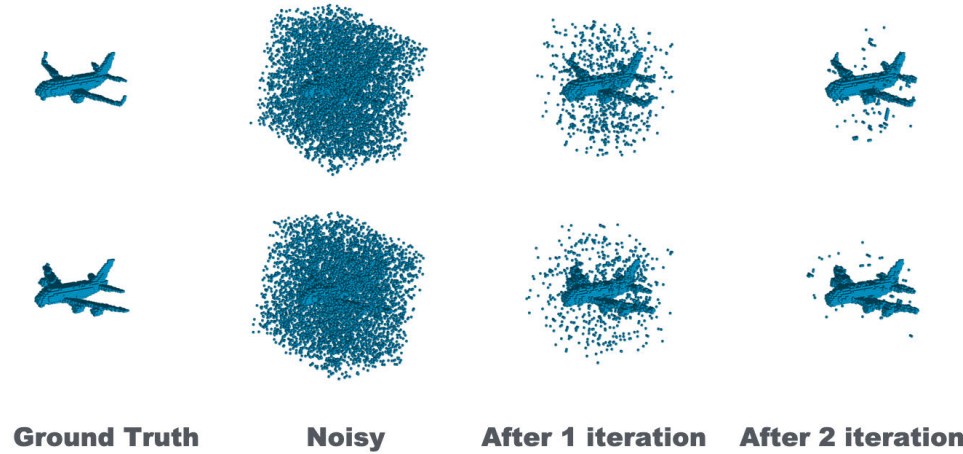

Figure 6: Result on Iterative Denoising

# E    NOISE MASKING STRATEGY

We found that adding random noise to the whole object is more challenging for LLM, thus leading to a more robust model. Random noises applied in our experiments include many outliers that put VP-LLM to the real test. They also include quantization and misalignment noises, typical of real data capture, which are considered easier here because, unlike outliers, they can be largely eliminated after discrete tokenization. We present some of the results using the strategy that adding more noises to the parts around and on the model, the result is better since it is a simpler task, see Fig. 7.

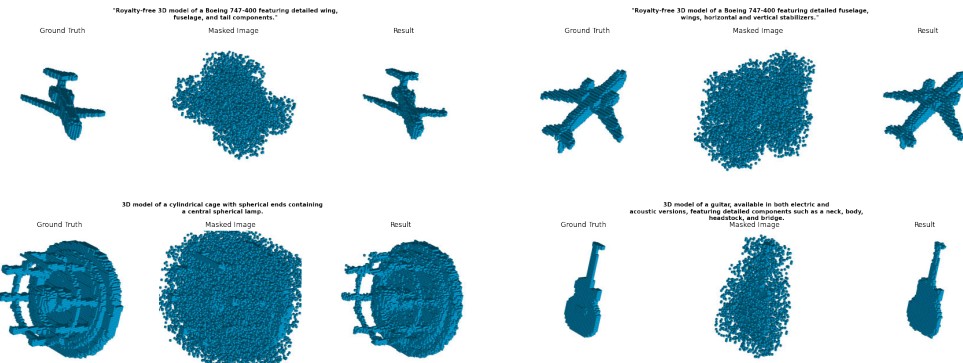

Figure 7: This figure demonstrates some results of another masking strategy, by which the original shape cannot be easily recognized. As suggested by reviewers, we add more noise around the object and gradually reduce the noise level when stepping away from the object. We can see the results are even better than those of uniform noise cases.

# F    MORE RESULTS

Figure 8 to 17 present more results of more categories, notice that all these results are inferenced from the same checkpoint as used in the experiment session.

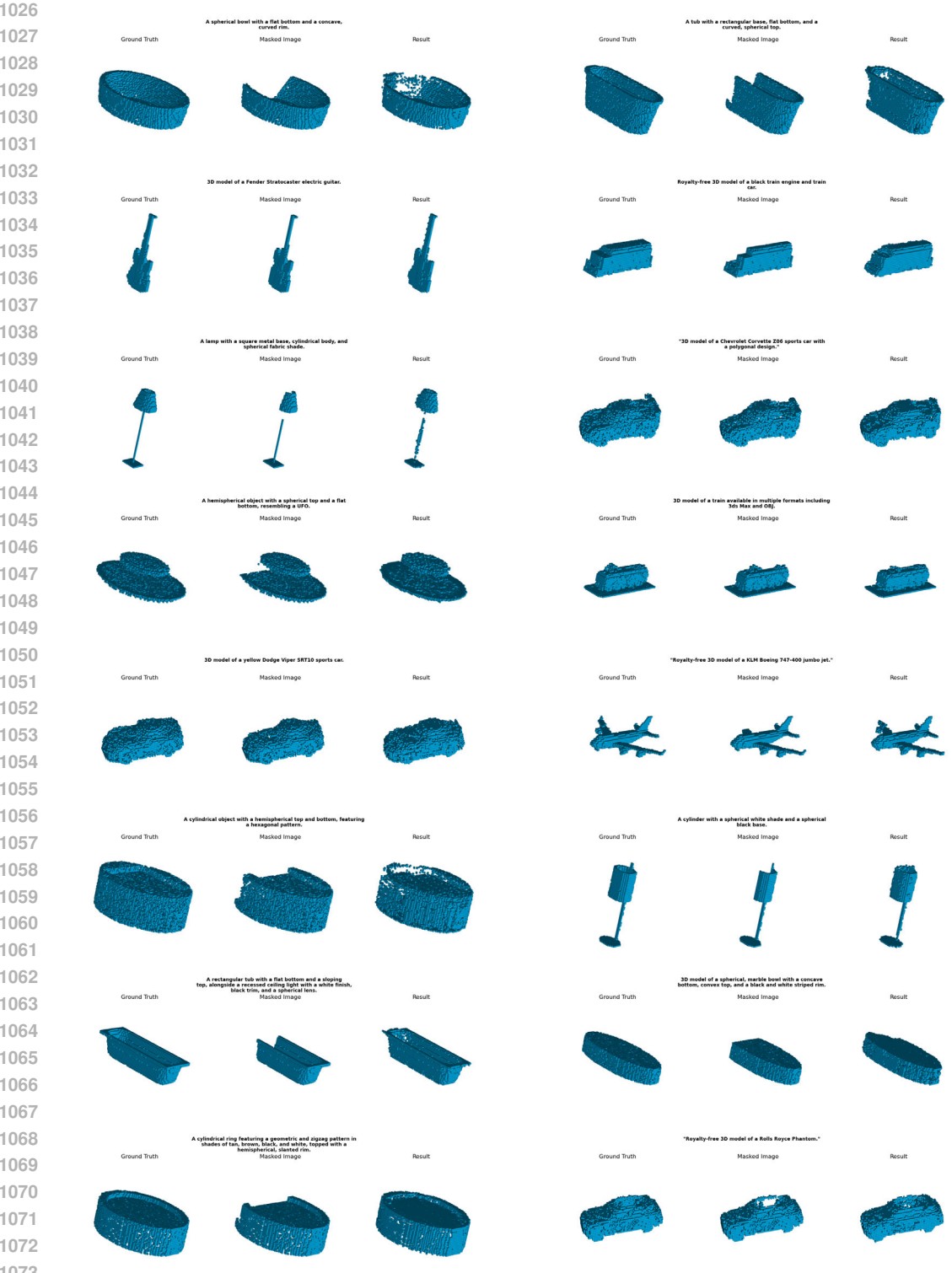

Figure 8: Results Seg20%

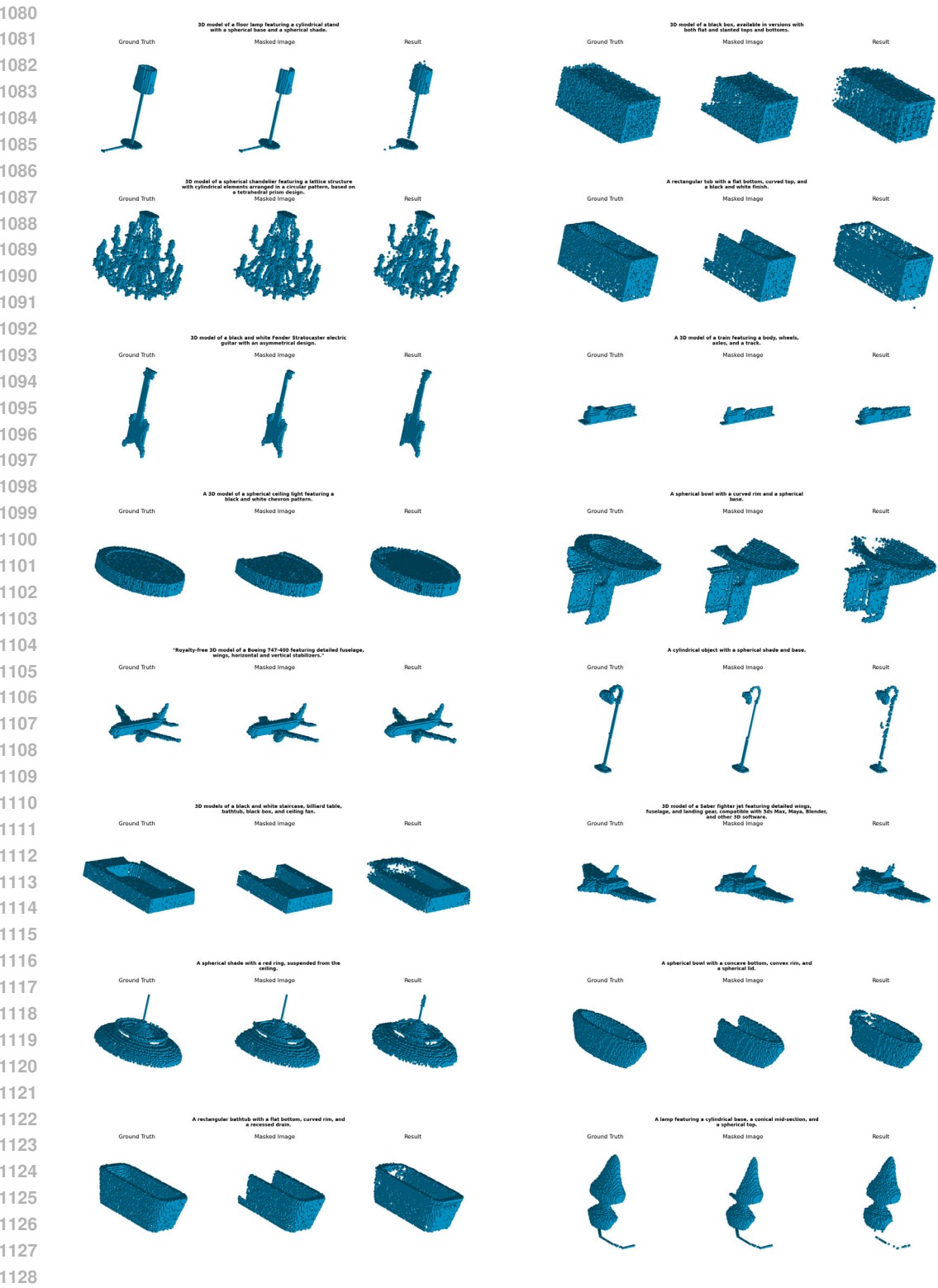

Figure 9: Results Seg20%

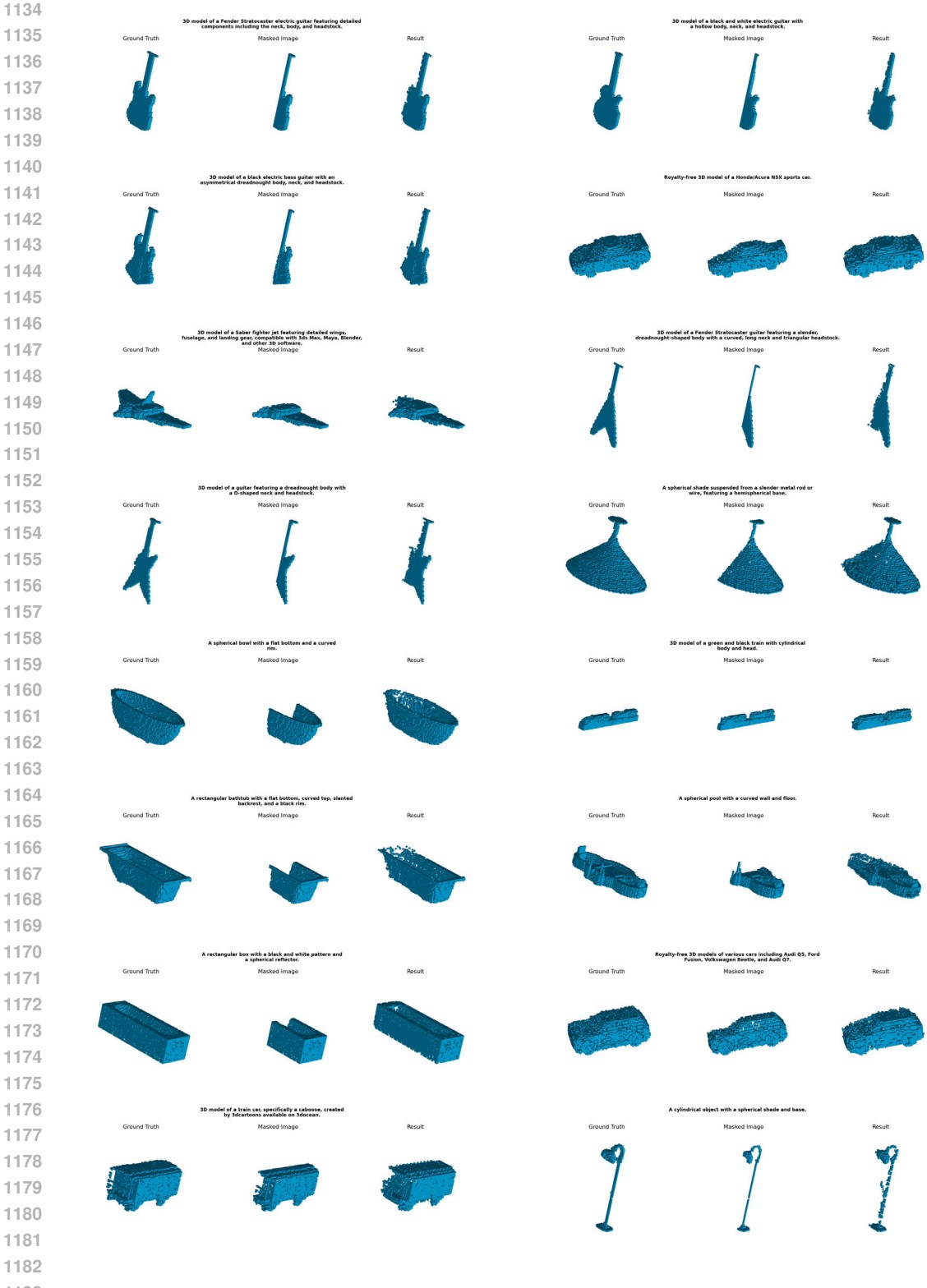

Figure 10: Results Seg50%

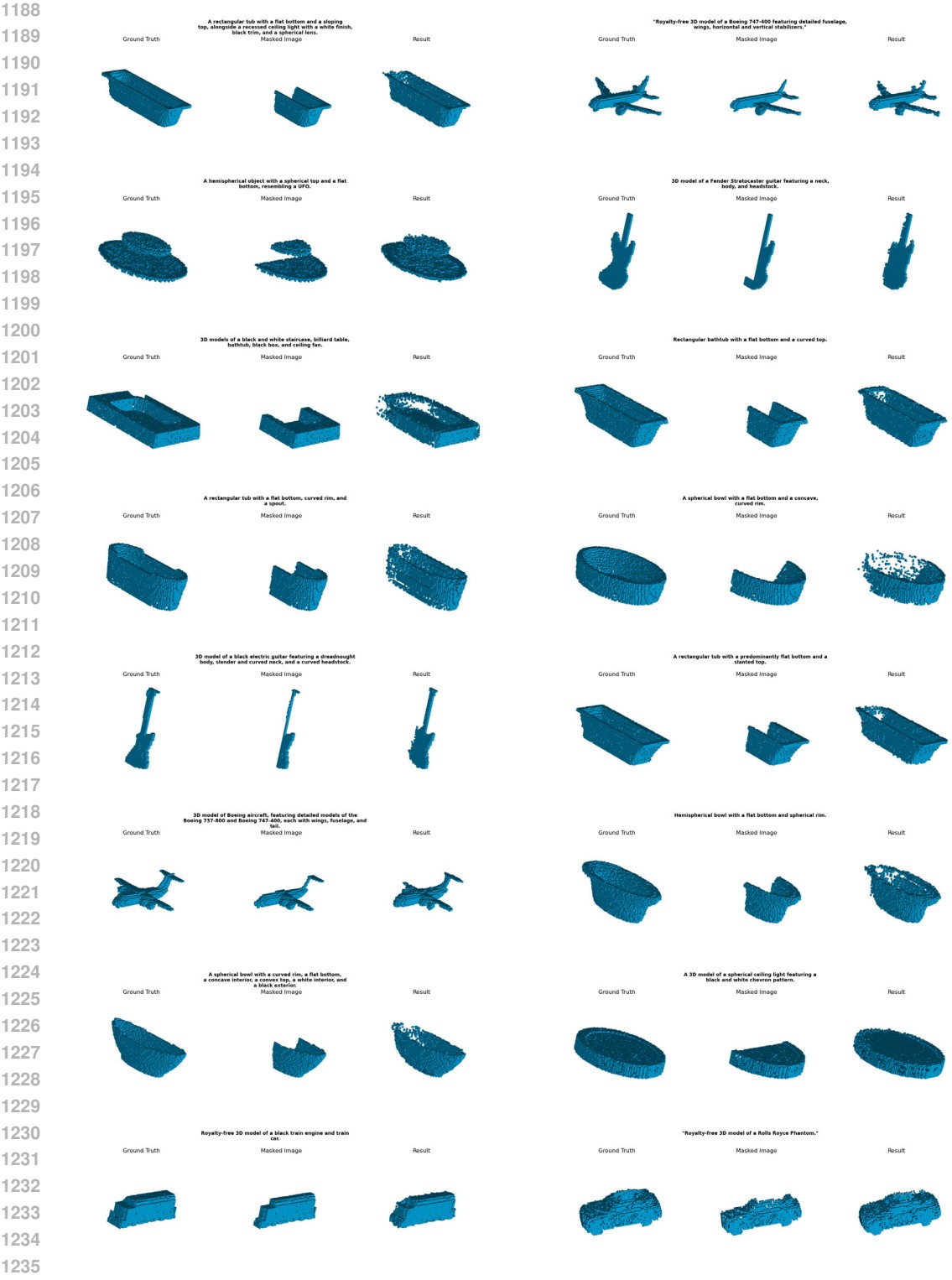

Figure 11: Results Seg50%

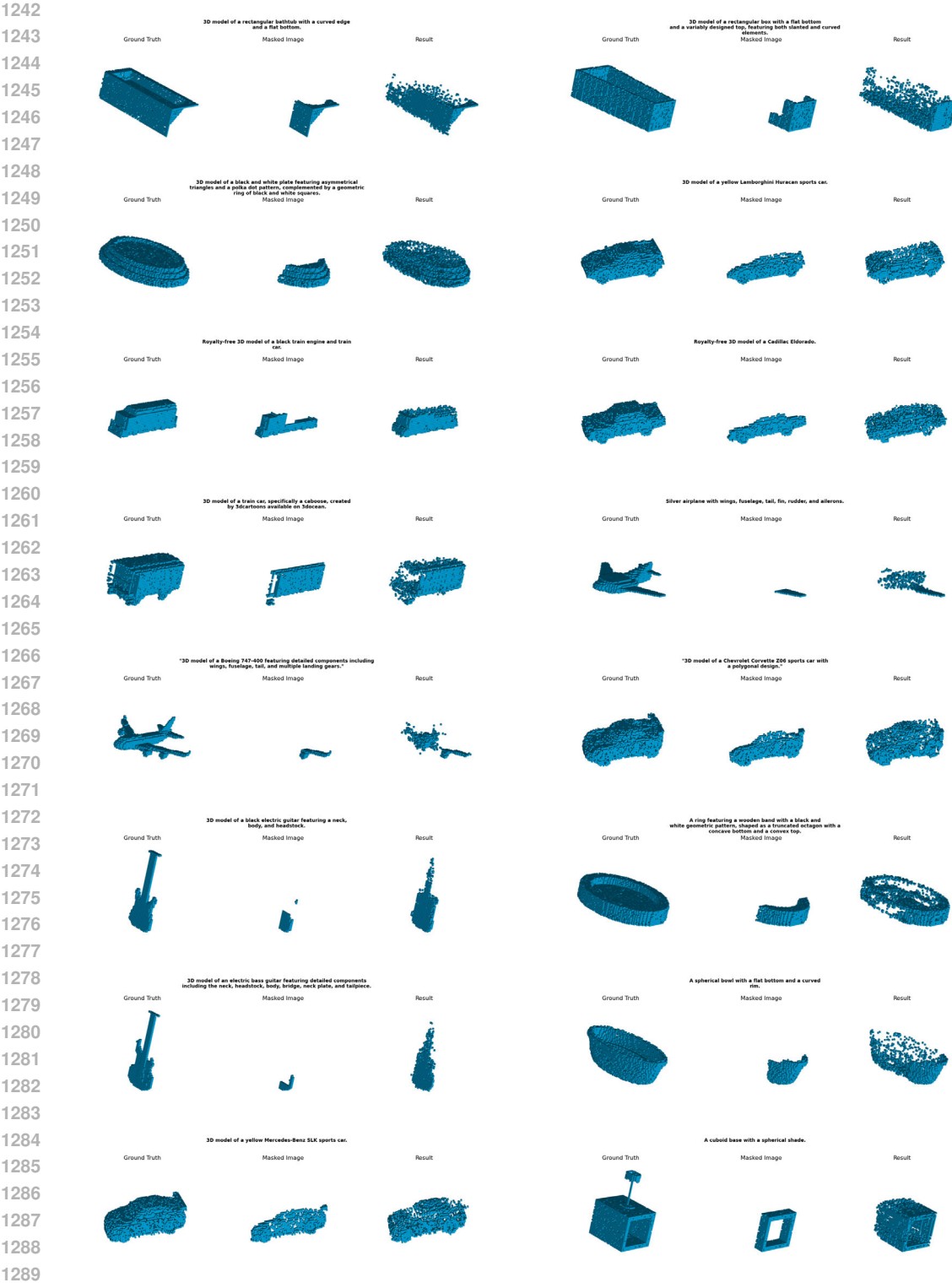

Figure 12: Results Seg80%

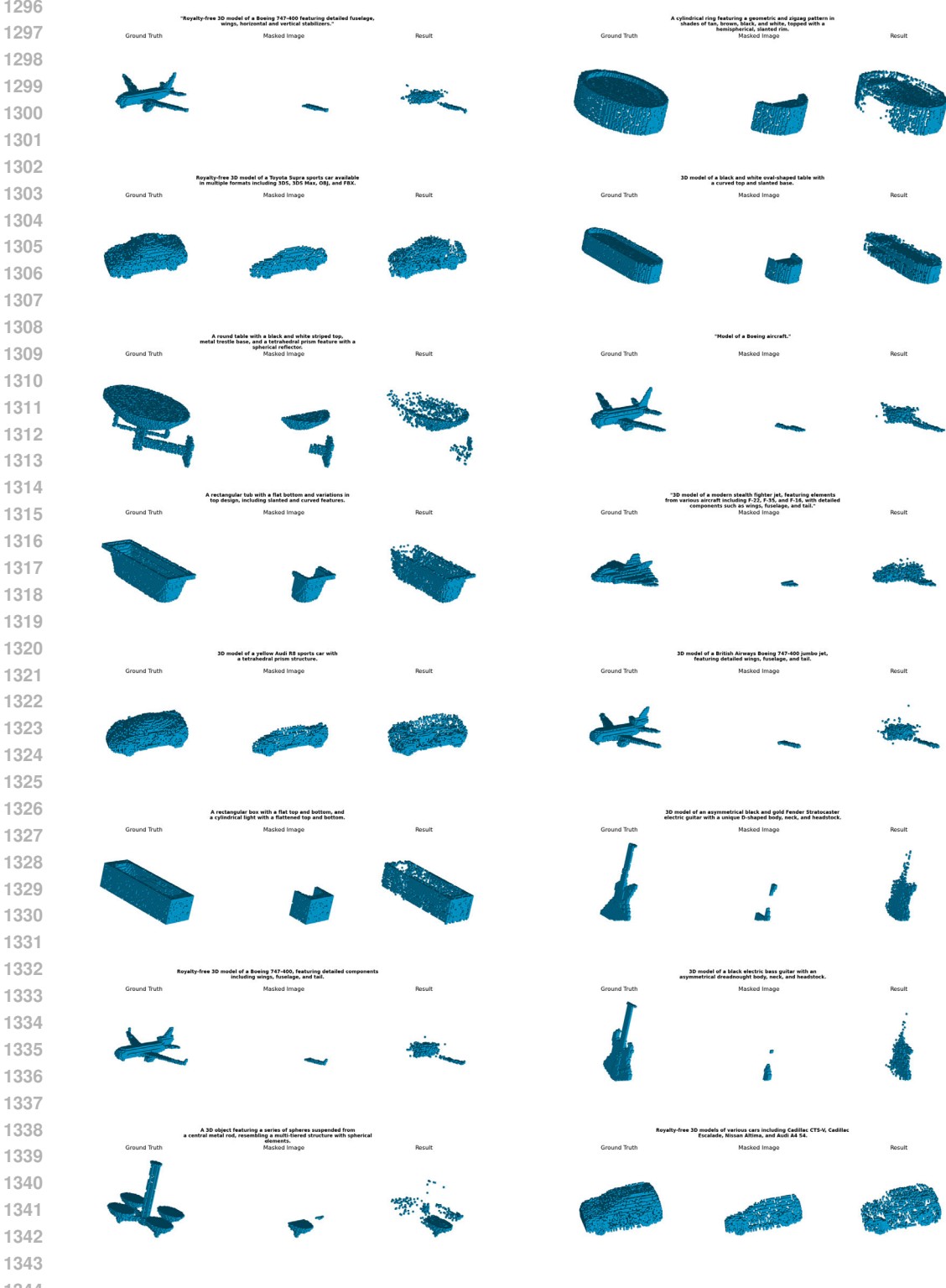

Figure 13: Results Seg80%

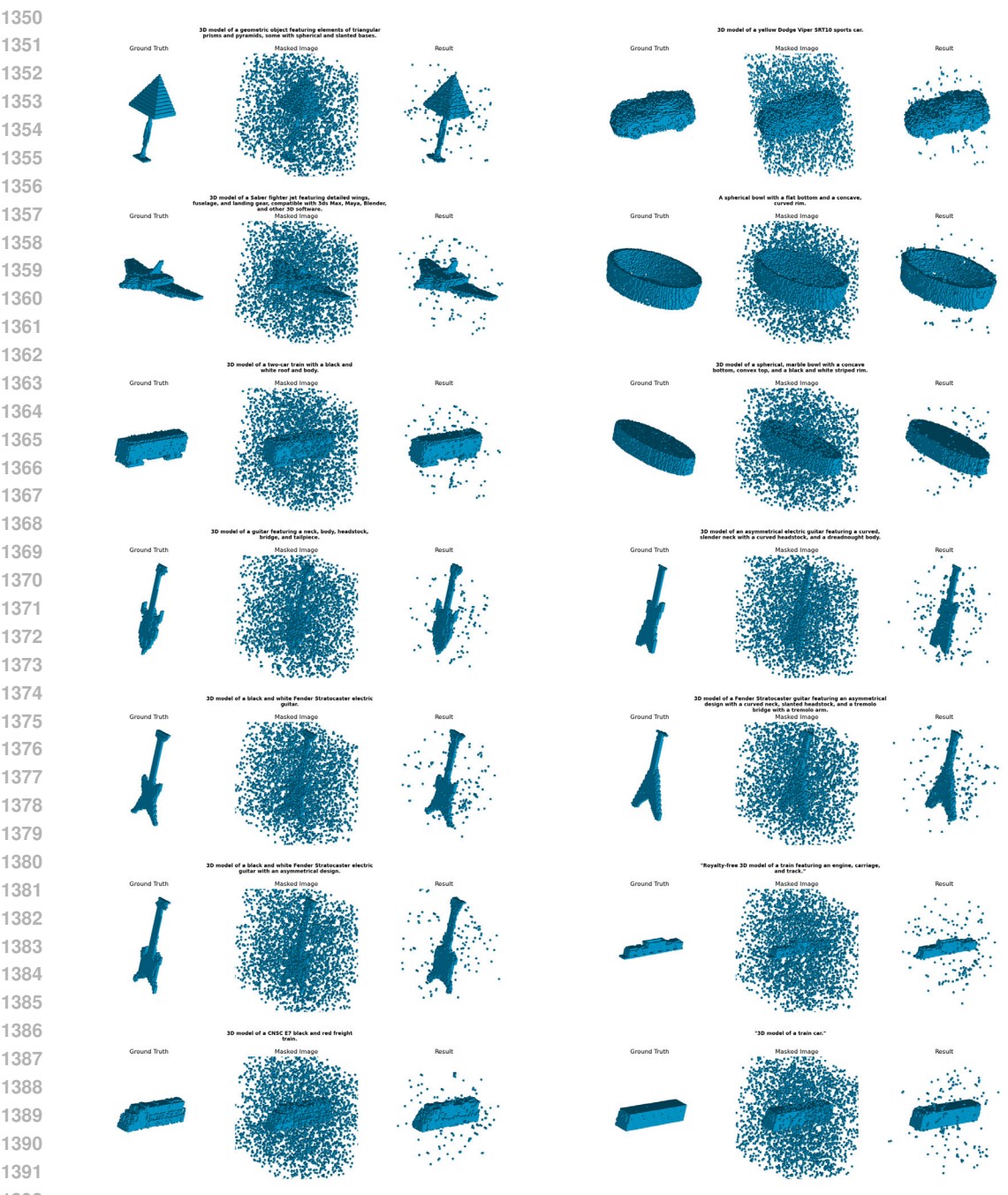

Figure 14: Results Noise1%

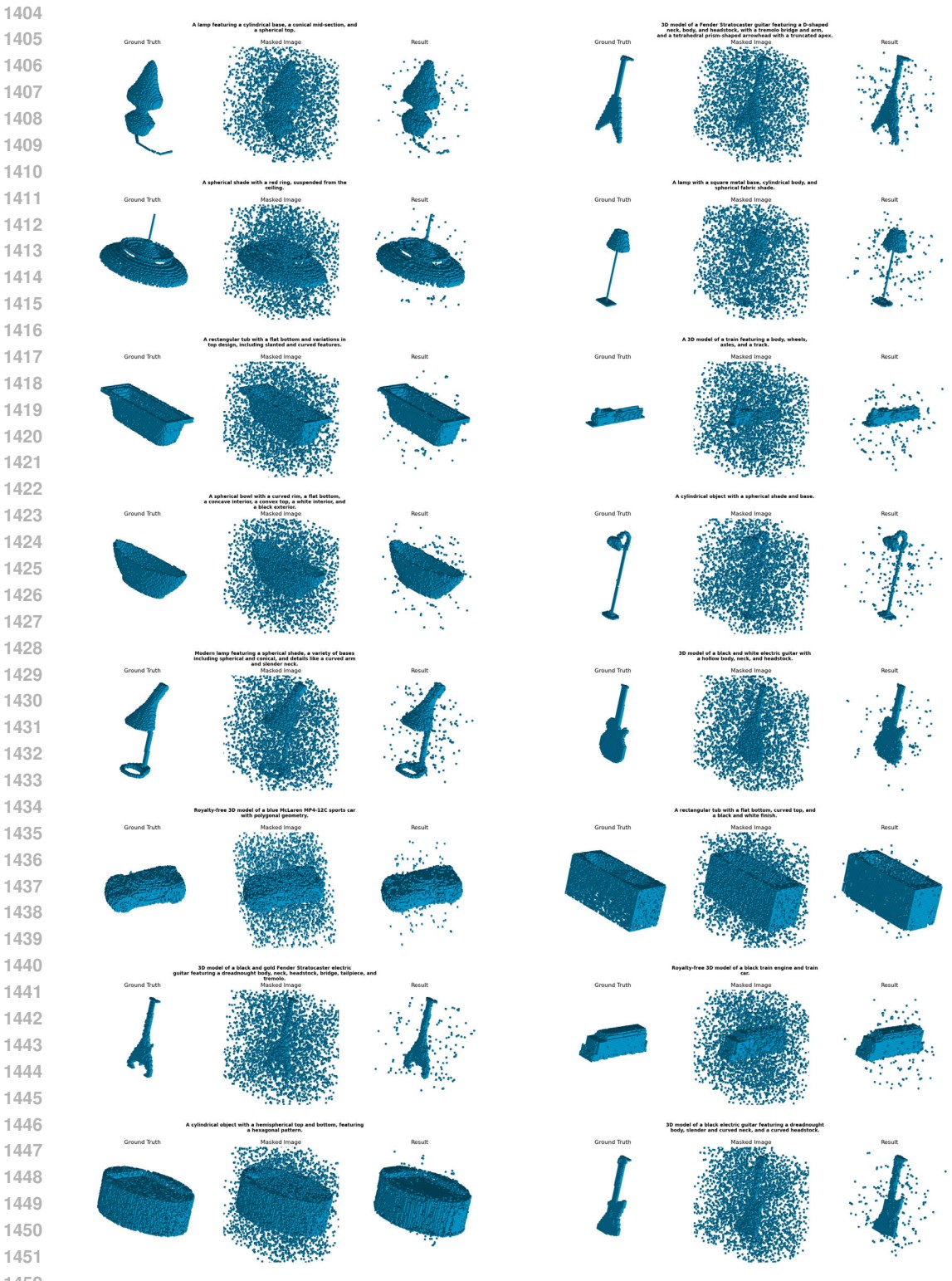

Figure 15: Results Noise1%

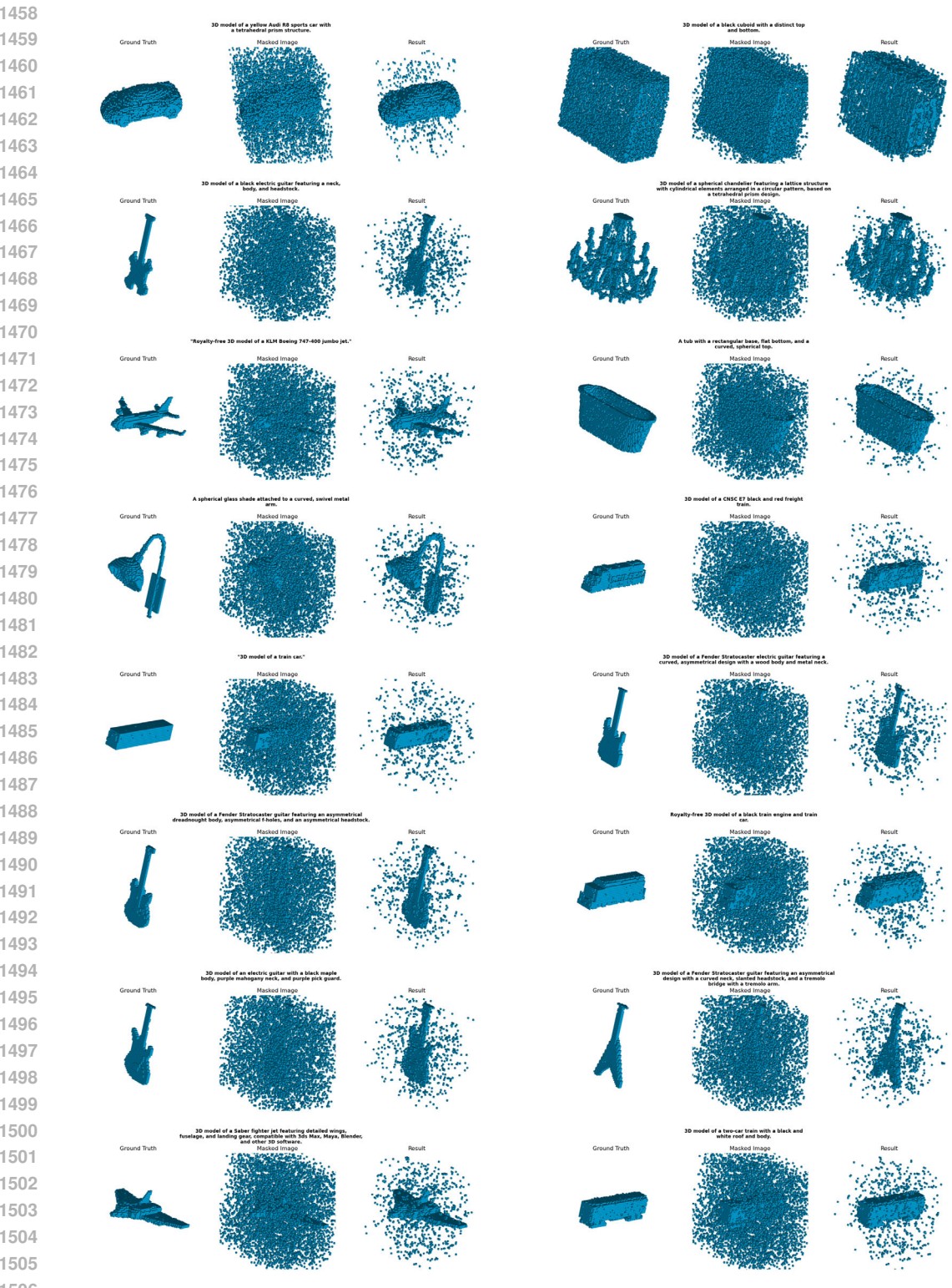

Figure 16: Results Noise2%

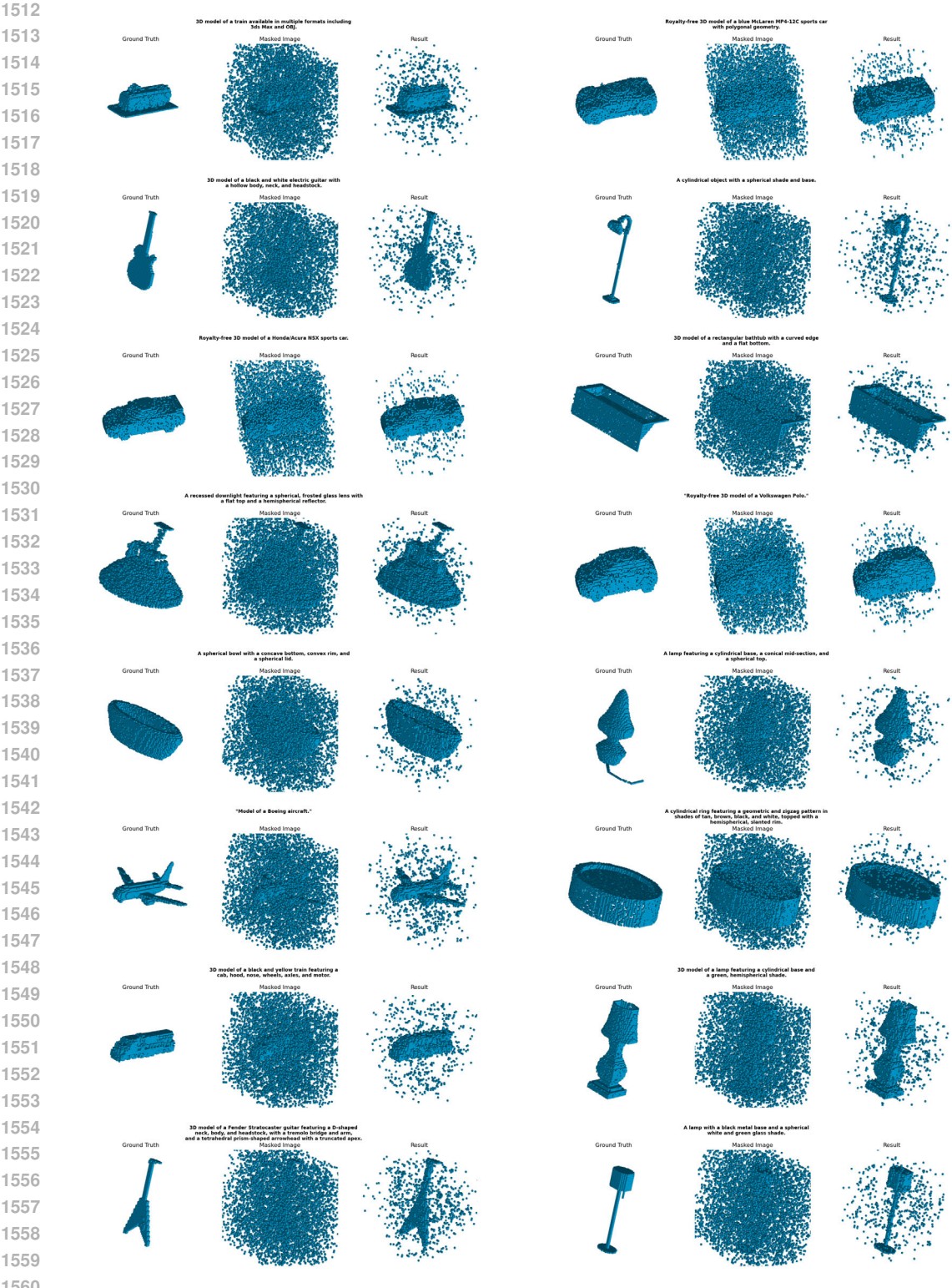

Figure 17: Results Noise2%

