# OpenReview forum: "VP-LLM: Text-Driven 3D Volume Completion with Large Language Models through Patchification"
_ICLR.cc/2025/Conference — Submitted to ICLR 2025_

### Official Review · Reviewer_5Mao · 2024-10-27

**Soundness:** 3
**Presentation:** 3
**Contribution:** 1
**Rating:** 3
**Confidence:** 4

**Summary:**

This paper introduces an LLM-based solution to shape completion. Given a 3D model and a text prompt, first, the incomplete model is projected to a sequence of latent vectors through a VAE. Then it is projected to the LLM latent space and concatenated with the encoded prompt. The LLM (+LORA) parses it and decodes a patched version of the complete model. Finally, a decoder reconstructs the final 3D model. The different components are trained, or fine-tuned, independently.
The capabilities of the model are shown on ShapeNet and compared with different alternative approaches.

**Strengths:**

The proposed approach shows it is possible to use LLM with 3D modalities through LORA. Furthermore, the proposed approach shows a way of interactively completing shapes, which has been underexplored so far. The denoising and completion tasks are quite important results of this manuscript.

**Weaknesses:**

Different points in the paper concern me. I have listed them below and I hope the authors can address these points in the rebuttal.

1. ShapeNet is known to have axis-aligned models, what happens when this is not the case?
2. The authors should present results on other datasets.
3. The shape completion and editing tasks are quite similar to shape editing when encompassing a text prompt. Such an application would make the paper much stronger. Has the author tested this application? In particular, the input shape would be a complete one, rather than incomplete.

These 3 points are those that drive my final score. I hope the authors can address them appropriately during the rebuttal period.

**Questions:**

Please see the weakness section.

---

### Official Review · Reviewer_gmoW · 2024-10-29

**Soundness:** 3
**Presentation:** 3
**Contribution:** 3
**Rating:** 5
**Confidence:** 3

**Summary:**

This paper presents VP-LLM for text-driven 3D completion. The authors divide a shape into smaller patches and encode each patch with a patch-wise variational autoencoder (VAE), allowing the LLM to process 3D data alongside textual instructions effectively. VP-LLM uses the ability of LLMs to complete 3D models and demonstrates good denoising capabilities. Experimental results show that VP-LLM outperforms existing models in 3D completion and denoising tasks.

**Strengths:**

+ The overall pipeline seems reasonable by combining existing techniques.
+ Leveraging LLM to complete 3D shapes is interesting. It would be interesting to investigate how large pre-trained models can benefit other modal tasks.
+ VP-LLM outperforms existing methods such as SDFusion and 3DQD, showing its effectiveness in 3D completion and denoising tasks. Compared with the baseline, VP-LLM can reduce more severe noise, while the baseline cannot do this.

**Weaknesses:**

- The shape rendering is of very low quality, making it hard to judge shape details.
- Although the paper emphasizes the advantages of LLM in understanding complex text instructions and long contexts, there is a lack of comparative analysis. The author did not explain the shortcomings of traditional methods (such as CLIP or BERT) in specific tasks, and in what specific aspects VP-LLM surpasses these methods. There is a lack of detailed discussion on why LLM is inherently superior to other multimodal models designed specifically for 3D tasks.
- The generalization ability of the method needs further evaluation. The data distribution of ShapeNet is different from the real world. It is interesting to see results on more complex datasets, e.g., Objaverse.
- Only two baseline methods were selected in the comparative experiment. The author can try to compare with point cloud-based methods， There are relatively more alternatives.
- The method requires the use of a decoder for completion, which may limit its generalization ability on more complex or different types of 3D data. However, the paper did not conduct an in-depth analysis of whether the output part would limit the model's capabilities.
- This paper conducted an ablation study on different LLM architectures and voxel resolutions but did not study the impact of other key hyperparameters (such as patch size and number of patches). Testing how these factors affect performance will help evaluate the robustness and flexibility of VP-LLM.

**Questions:**

+ Does the method have an advantage in 3D generation? I did not see any obstacle in extending it to 3D generation. Why not test in this task?

+ The author can compare the denoising part with some other methods, such as Point-DAE, and Point-E. The comparison method mentioned in the paper does not support it. Comparing with some other models that can reduce noise can further illustrate the effectiveness of the method.

**Details Of Ethics Concerns:**

NIL

---

### Official Review · Reviewer_TDXM · 2024-11-02

**Soundness:** 3
**Presentation:** 3
**Contribution:** 3
**Rating:** 5
**Confidence:** 3

**Summary:**

The paper introduces a novel framework that enables large language models (LLMs) to perform 3D shape completion. The framework processes incomplete or noisy 3D volumes along with textual instructions to interactively generate complete or refined models. In this approach, the 3D volume is divided into patches and sequentialized for input into a transformer backbone, which can remain frozen during training. This work is the first to leverage pretrained LLMs for 3D shape completion, with experimental results demonstrating the potential of this design.

**Strengths:**

* Overall, the idea of leveraging pretrained LLMs for the 3D completion task is quite novel. Previous studies typically concentrate on integrating LLMs for 3D perception tasks, whereas this work is the first to utilize LLMs for 3D generative tasks.

* The paper is well-organized and clearly presented, enhancing its readability and comprehension.

**Weaknesses:**

The primary concern is that the demonstrated task is overly simplified, and the experimental results are not sufficiently convincing:
- The resolution of 64 is too low to capture detailed representations. Such low-resolution volumes fail to accurately reflect the textual descriptions (e.g., "perpendicular wings" and "solar plane" in Table 4), thereby diminishing the motivation for instruction-based generation.
- All presented examples appear to involve simple shapes, such as airplanes, cars, and lamps. ShapeNet includes more complex shapes, such as tables and chairs, which would better showcase the framework's performance and robustness.

**Questions:**

- The performance of the compared baselines appears significantly poorer than reported in their original papers (e.g., SDFusion results mostly show disconnected points, whereas smoother volumes would be expected). It is unclear what causes this discrepancy. Converting the SDF to an isosurface mesh, rather than visualizing voxels directly, might provide a more accurate comparison.
- Although the author acknowledges the low resolution of the volumes, the current results remain suboptimal. Many studies have focused on better compression and generation of SDF/occupancy-based volumes, such as Michelangelo [1] and 3DShape2VecSet [2]. These works propose encoding high-resolution 3D volumes into compact sequential codes using a VAE, which could be better alternatives.
- It is unclear why the projection layers utilize independent MLP clusters instead of a more commonly used shared MLP head.

[1] Conditional 3D Shape Generation based on Shape-Image-Text Aligned Latent Representation
[2] 3DShape2VecSet: A 3D Shape Representation for Neural Fields and Generative Diffusion Models

---

### Official Review · Reviewer_6h6x · 2024-11-04

**Soundness:** 3
**Presentation:** 3
**Contribution:** 3
**Rating:** 6
**Confidence:** 3

**Summary:**

The paper presents VP-LLM, which leverages Large Language Models (LLMs) to achieve text-guided 3D volume completion. Traditional methods for 3D completion, typically reliant on diffusion models, face scalability limitations, particularly in handling high-resolution voxel grids and understanding complex textual descriptions. VP-LLM addresses these issues by introducing a "patchification" technique, which segments the 3D volume into small patches. These patches are encoded independently using a Variational Autoencoder (VAE) and then mapped to the LLM’s embedding space, allowing for precise text-guided completion.

**Strengths:**

1. The paper introduces an innovative approach by leveraging Large Language Models (LLMs) for 3D volume completion guided by textual prompts. This novel integration of patchification with LLMs addresses the scalability issues seen in traditional voxel-based approaches, enabling finer control through text guidance.
2. The methodology is well-grounded, with detailed explanations of the VP-LLM model architecture, including the patchification process, VAE integration, and LLM fine-tuning steps. The use of quantitative metrics (Chamfer Distance and CLIP-s score) and comparisons against state-of-the-art models like SDFusion and 3DQD lend credibility to the results.
3. This work is significant for 3D computer vision, offering a scalable alternative to diffusion-based methods with increased flexibility through text control.

**Weaknesses:**

1. The comparison is only conducted on the Airplane and Car dataset, which is not aligned with the baseline SDFusion. It is highly recommended to provide more comparison results on the chair, table, or house for a fair comparison.
2. The model only supports text description as the condition. However, the baseline SDFuion enables conditional generation with multiple modalities like text description and RGB image. The image guidance is a more user-friendly way so missing this condition will limit the application capability of the model.
3. The experiments primarily use ShapeNet data and Cap3D-generated captions, which may limit the generalizability of results. Evaluating additional datasets or real-world 3D captures could strengthen the argument for VP-LLM’s applicability to diverse scenarios.
4. Compared with diffusion-based methods, how is the generation diversity of VP-LLM?

**Questions:**

1. What about the performance of VP-LLM if the text description is quite different from the training set generated by Cap3D?
2. Since the authors claim the model can perform completion tasks at a larger scale, why not show the 3D completion results for real-world scanned objects?
3. Why choose separate MLPs to process and generate each patch? How to deal with the huge memory cost if the number of patches increases?
4. Why not use the MLLM to support multi-modal conditions?

---

### Meta-Review · Area_Chair_xusg · 2024-12-17

**Metareview:**

The paper introduces the VP-LLM, which leverages large language models (LLMs) for text-guided 3D volume completion. Several notable limitations are raised. The results were primarily based on ShapeNet with limited comparisons to baseline methods. The low-resolution outputs weaken the claims of scalability and practical use cases. The lack of multi-modal conditioning (e.g., image guidance) and design choices further limits its novelty. I’d recommend the authors to consider all these and further improve the work.

**Additional Comments On Reviewer Discussion:**

The authors did not provide any rebuttal and no discussion conducted.

---

### Decision · Program_Chairs · 2025-01-22

Reject